# New Hydroxydecanoic Acid Derivatives Produced by an Endophytic Yeast *Aureobasidium pullulans* AJF1 from Flowers of *Aconitum carmichaeli*

**DOI:** 10.3390/molecules24224051

**Published:** 2019-11-08

**Authors:** Hyun Gyu Choi, Jung Wha Kim, Hyukjae Choi, Ki Sung Kang, Sang Hee Shim

**Affiliations:** 1College of Pharmacy, Duksung Women’s University, Seoul 01369, Korea; chg---@hanmail.net (H.G.C.); jwkim7317@gmail.com (J.W.K.); 2College of Pharmacy, Yeungnam University, Gyeongsan 38541, Korea; h5choi@yu.ac.kr; 3College of Korean Medicine, Gachon University, Seongnam 13120, Korea; kkang@gachon.ac.kr

**Keywords:** endophyte, *Aureobasidium pullulans*, hydroxydecanoic acid, *Aconitum carmichaeli*

## Abstract

Endophytes have been recognized as a source for structurally novel and biologically active secondary metabolites. Among the host plants for endophytes, some medicinal plants that produce pharmaceuticals have been reported to carry endophytes, which could also produce bioactive secondary metabolites. In this study, the medicinal plant *Aconitum carmichaeli* was selected as a potential source for endophytes. An endophytic microorganism, *Aureobasidium pullulans* AJF1, harbored in the flower of *Aconitum carmichaeli*, was cultured on a large scale and extracted with an organic solvent. Extensive chemical investigation of the extracts resulted in isolation of three lipid type compounds (**1**–**3**), which were identified to be (3*R*,5*R*)-3,5-dihydroxydecanoic acid (**1**), (3*R*,5*R*)-3-(((3*R*,5*R*)-3,5-dihydroxydecanoyl)oxy)-5-hydroxydecanoic acid (**2**), and (3*R*,5*R*)-3-(((3*R*,5*R*)-5-(((3*R*,5*R*)-3,5-dihydroxydecanoyl)oxy)-3-hydroxydecanoyl)oxy)-5-hydroxydecanoic acid (**3**) by chemical methods in combination with spectral analysis. Compounds **2** and **3** had new structures. Absolute configurations of the isolated compounds (**1**–**3**) were established using modified Mosher’s method together with analysis of NMR data for their acetonide derivatives. All the isolates (**1**–**3**) were evaluated for antibiotic activities against *Escherichia coli*, *Staphylococcus aureus*, *Bacillus subtilis*, *Pseudomonas aeruginosa,* and their cytotoxicities against MCF-7 cancer cells. Unfortunately, they showed low antibiotic activities and cytotoxic activities.

## 1. Introduction

Endophytes spend all or part of their lives colonizing inside of healthy tissues of a host plant. They cause no apparent disease symptoms in host plant, and rather give abilities including tolerance to biotic and abiotic stress, nutrient acquisition, and plant growth promotion to the host plant [1]. Various types of secondary metabolites with significant bioactivities have been produced by endophytes as a result of interaction with host plant [2]. Biosynthesis of many metabolites has been considered to be caused by a result of environmental signals such as insects and plant pathogen [3]. In a search for endophytes to produce secondary metabolites with intriguing structures and potent bioactivities, an endophytic yeast *Aureobasidium pullulans* AJF1 was isolated from flowers of *Aconitum carmichaeli*. Some studies have revealed that *A. pullulans* produce various types of metabolites such as polysaccharides, mannitol oils, poly malic acid, 2-propylacrylic acid, 8,9-dihydroxy-2-methyl-4*H*,5*H*-pyrano[3,2-c]chromon-4-one, 2-methylenesuccinic acid, hexane-1,2,3,5,6-hexol, etc. [4,5,6,7]. Roots of *Aconitum carmichaeli* Debeaux have been treated as one of the important crude drugs in oriental medicine due to its potent activities to treat shock caused by acute and chronic heart failure and low blood pressure as well as toxicities [8]. Even though a broad spectrum of research on the secondary metabolites produced by endophytes from medicinal plants has been carried out since discovery of paclitaxel production by the endophytic fungus, *Taxomyces andreanae* [9,10], only a few studies have been conducted on the endophytes isolated from *A. carmichaeli* [11,12,13]. Herein, three metabolites (**1**–**3**) were isolated from cultures of the endophytic yeast *A. pullulans* AJF1 and investigated for their activities (Figure 1). They were all hydroxy fatty acid derivatives and compound **1**, 3,5-dihydroxydecanoic acid, was previously reported to be one of main components of extracellular lipids produced by *Aureobasidium* sp. [14]. However, its absolute as well as relative stereochemistry was elucidated for the first time in this study.

## 2. Results and Discussion

Three hydroxy fatty acid derivatives were isolated from cultures of an endophytic yeast strain of *Aureobasidium pullulans* AJF1. Mass cultivation of the strain, extraction, followed by a series of chromatographic methods led to the isolation of three metabolites (**1**–**3**). Their chemical structures were determined to be (3*R*,5*R*)-3,5-dihydroxydecanoic acid (**1**), (3*R*,5*R*)-3-(((3*R*,5*R*)-3,5-dihydroxydecanoyl)oxy)-5-hydroxydecanoic acid (**2**), and (3*R*,5*R*)-3-(((3*R*,5*R*)-5-(((3*R*,5*R*)-3,5-dihydroxydecanoyl)oxy)-3-hydroxydecanoyl)oxy)-5-hydroxydecanoic acid (**3**) by spectroscopic data analyses such as NMR and MS together with a variety of chemical reactions.

Compound **1** was isolated as colorless oil. The molecular formula was determined to be C_10_H_20_O_4_ from positive HRFABMS at *m*/*z* 187.1331 [M − H_2_O + H]^+^ (Calcd for C_10_H_19_O_3_, 187.1329). ^1^H-NMR of compound **1** showed two oxygenated methines at δ_H_ 4.66 (dtd, *J* = 11.3, 5.3, 3.1 Hz) and 4.33 (quint, *J* = 3.7 Hz), a terminal methyl at δ_H_ 0.85 (t, *J* = 6.9 Hz), together with several methylene signals. Its ^13^C-NMR spectrum exhibited the presence of ten carbons including a carboxylic carbon at δ_C_ = 171.2, two oxymethine carbons at δ_C_ 76.1 and 62.5, six methylene carbons at δ_C_ 38.5, 35.7, 35.4, 31.5, 24.5, and 22.5, and a terminal methyl carbon at δ_C_ 13.9, suggesting that compound **1** was a dihydroxy decanoic acid. The positions of two hydroxyl groups were confirmed by 2D-NMR data including ^1^H-^1^H COSY, HSQC, and HMBC. A C-CH_2_-CH(O)-CH_2_-CH(O) spin system was obtained by analysis of its ^1^H-^1^H COSY spectrum (Figure 2). HMBC correlations of the methylene protons at δ_H_ 2.67 and 2.58 with the carboxylic carbon (δ_C_ 171.2), oxymethine carbon (δ_C_ 62.5), and a methylene carbon (δ_C_ 35.7) indicated that the spin system was directly linked to carboxylic acid, that is, 3,5-dihydroxy decanoic acid (Figure 1). This structure was further confirmed by HMBC correlations of the oxymethine proton at δ_H_ 4.33 with the carboxylic carbon, two methylene carbons (δ_C_ 38.5 and 35.7), and the oxymethine carbon (δ_C_ 76.1).

Compound **2** was isolated as colorless oil. Its molecular formula was established to be C_20_H_38_O_7_ on the basis of positive HRFABMS at *m*/*z* 373.2590 [M − H_2_O + H]^+^ (Calcd for C_20_H_37_O_6_, 373.2585). ^1^H and ^13^C-NMR spectra of compound **2** were similar to those for compound **1**, differing in that compound **2** has ten more carbons including two more oxymethine groups and one more carboxylic acid than **1** (Table 1). Two units of C-CH_2_-CH(O)-CH_2_-CH(O) spin system were observed in its ^1^H-^1^H COSY spectrum (Figure 1). Thus, it was presumed that two units of dihydroxy fatty acid might be esterified to form compound **2**, which was supported by the downfield shifted oxymethine proton signal (δ_H_ 5.27). The connection of two units was verified by HMBC correlations. HMBC correlations of the oxymethine proton at δ_H_ 5.27 with a carboxylic acid carbon (δ_C_ 169.2), a carboxylic ester carbon (δ_C_ 171.1), and two methylene carbons (δ_C_ 35.3 and 32.7) supported that one dihydroxy fatty acid moiety was esterified to the other dihydroxy fatty acid moiety at the C-3 position (Figure 1). The possibility that two dihydroxy fatty acids might have different carbon numbers was not excluded. However, the presence of two decanoic acid moieties was confirmed by ten carbon atoms observed in the ^13^C-NMR spectrum for hydrolysate of compound **2**. Thus the planar structure of compound **2** was elucidated to be 3-((3,5-dihydroxydecanoyl)oxy)-5-hydroxydecanoic acid.

Compound **3** was also isolated as colorless oil. Its molecular formula was established to be C_30_H_56_O_10_ from the positive HRFABMS at *m*/*z* 581.3657 [M – H_2_O + Na]^+^ (Calcd for C_30_H_54_O_9_Na, 581.3660). The ^1^H and ^13^C-NMR of compound **3** were also similar to those for compound **2** (Table 1). As in the cases of compounds **1** and **2**, ten more carbons including two additional oxymethines (δ_C_ 69.8 and 72.4), one more carboxylic ester carbon (δ_C_ 172.3), six methylene carbons (δ_C_ 42.7, 42.4, 37.9, 25.0, 31.5, 22.5), and a terminal methyl group (δ_C_ 14.0) appeared compared with compound **2**, suggesting that three dihydroxy fatty acid moieties were linked together through ester linkage. The positions of the esterification were confirmed by HMBC correlations (Figure 2). HMBC correlations of the oxymethine H-3 at δ_H_ 5.29 with a carboxylic acid carbon C-1 at δ_C_ 171.5 and a carboxylic ester carbon C-1′ at δ_C_ 169.0) indicated that one dihydroxy fatty acid was esterified at the C-3 position of another dihydroxy fatty acid moiety. In addition, HMBC correlations of the oxymethine H-5′ at δ_H_ 5.04 with the last carboxylic ester carbon at δ_C_ 172.3 and the oxymethine carcon C-3′ at δ_C_ 66.3 indicated that the last dihydroxy fatty acid moiety was esterified to C-5′ position of one of the dihydroxy fatty acid moiety. As shown in compound **2**, the ^13^C-NMR spectrum of hydrolysate of compound **3** indicated the presence of three dihydroxy decanoic acids, suggesting the planar structure of compound **3**.

Relative configurations of compounds **1**–**3** were determined on the basis of semi-synthetic structure modification and their spectroscopic data analyses. The relative configuration of the acyclic 1,3-diols in compounds **1**–**3** were assigned by using the [^13^C]acetonide method [15]. Compound **2** was hydrolyzed with 1 N HCl in MeOH to yield the 3,5-dihydroxydecanoic acid methyl ester (**2a**) and 3,5-dihydroxydecanoic acid (**2b**) as depicted in Figure 3. Analogous treatment of compound **3** also afforded the 3,5-dihydroxydecanoic acid methyl ester (**3a**) and 3,5-dihydroxydecanoic acid (**3b**). Treatments of **1**, **2a**, **2b**, **3a**, and **3b** with 2,2-dimethoxypropane and pyridinium-*p*-toluensulfonate in methanol afforded the acetone ketal (**1a**, **2c**, and **3c**) at their 1,3-diol positions (Figure 3). Two methyl resonances observed at δ_C_ 30.1 and 19.7 for the acetone ketals suggested the relative configurations of the 1,3-diol to be in *syn* configurations. Modified Mosher’s method was employed to determine the absolute configurations of hydroxyl groups in these compounds. Treatment of **1** with (*R*) and (*S*)-MTPA-Cl and a catalytic amount of DMAP (4-dimethylaminopyridine) in pyridine-*d*_5_ afforded its (*S*) and (*R*)-MTPA ester, respectively. Low field shifted proton resonance at δ_H_ 4.33 for H-3 implied that the (*R*) and (*S*)-MTPA-Cl reacted only with the hydroxyl group at the C-3 position not that at C-5. Analysis of ^1^H-NMR chemical shift differences (∆δ*_S_*_-*R*_) between (*R*) and (*S*) Mosher’s ester of **1** revealed the absolute configuration of C-3 and C-5 to be *R* and *R*, respectively, as shown in Figure 2. Consequently, the structure of **1** was determined to be (3*R*,5*R*)-3,5-dihydroxydecanoic acid. Likewise, compound **2** was derivatized to its (*R*) and (*S*)-MTPA ester using (*S*) and (*R*)-MTPA-Cl, respectively, in which MTPA groups were confirmed to be attached to all the hydroxyl groups of compound **2** by chemical shifts of oxymethine protons. Analysis of ^1^H-NMR chemical shift differences (∆δ*_S_*_-*R*_) between (*R*) and (*S*) tri-Mosher’s ester of **2** revealed that C-3, 5, 3′, and 5′ were all in *R* configurations as shown. Thus, compound **2** was identified to be (3*R*,5*R*)-3-(((3*R*,5*R*)-3,5-dih ydroxydecanoyl)oxy)-5-hydroxydecanoic acid. Determination of the absolute configurations of **3** using Mosher’s method was quite challenging since the proton resonances for methylene groups neighboring oxymethines were too difficult to be distinguished from others since many proton resonances were overlapped [16]. Nevertheless, Mosher’s esterification of the hydrolysates of compound **3** allowed the determination of the absolute stereochemistry, since compound **3** was composed of three units of 3,5-dihydroxydecanoic acid moiety. To obtain each monomer of compound **3**, the acetonide derivative 3C was treated with pyridinium-*p*-toluensulfonate at 40 °C in MeOH to afford **3d**. Analysis of ^1^H-NMR chemical shift differences (Δδ*_S_*_-*R*_) of (*S*) and (*R*) esters of 3d elucidated the absolute configuration of C-3 and 5 to be *R* and *R* configurations, respectively (Figure 4). Finally, chemical structure of **3** was determined as (3*R*,5*R*)-3-(((3*R*,5*R*)-5-(((3*R*,5*R*)-3,5-dihydroxydecanoyl)oxy)-3-hydroxydecanoyl)oxy)-5-hydroxydecanoic acid.

Compound **1**, (3*R*,5*R*)-3,5-dihydroxydecanoic acid was previously reported as a not only synthetic intermediate but also natural product [14,17]. 3,5-Dihydroxydecanoic acid was reported to be a main component of the lipophilic moieties in lipid produced by *Auresobasidium* sp. [14]. However, its relative and absolute stereochemistry was established in this study for the first time. The (3*R*,5*R*)-3,5-dihydroxydecanoic acid moiety is extremely rare in nature. As one example of natural compounds to bear this moiety, exophilin A was isolated from a marine microorganism *Exophiala pisciphila* [18]. Exophilin A is also a trimer of the (3*R*,5*R*)-3,5-dihydroxydecanoic acid with the ester linkages at 5-OH and 5′-OH while the esters in compound **3** at 3 and 5′. The relative and absolute stereochemistry of exophilin A were elucidated by comparison of the optical rotation value for its hydrolysate with that for (3*R*,5*R*)-3-hydroxy-5-decanolide [18]. In this study, the acetonide method, Mosher’s method, and acid hydrolysis together with NMR analysis were employed to elucidated the stereochemistry of the isolated compounds. All the isolates (**1**–**3**) were evaluated for antibiotic activities against *Escherichia coli*, *Staphylococcus aureus*, *Bacillus subtilis*, *Pseudomonas aeruginosa,* and their cytotoxicities against MCF-7 cancer cells. Unfortunately, they showed low antibiotic activeities and cytotoxic activities. While exophilin A was reported to be active against *Staphylococcus aureus* with an MIC of 50 μg/mL, compound **3** did not show antibacterial activities even at 128 μg/mL.

## 3. Materials and Methods

### 3.1. General Experimental Procedures

The high-resolution fast atom bombardment mass spectrometry (HRFABMS) data were obtained on gas chromatography high resolution mass spectrometer (JMS-700, Jeol, Tokyo, Japan). The nuclear magnetic resonance (NMR) spectra were acquired with a 300 Ultra shield spectrometer (^1^H, 300 MHz; ^13^C, 75 MHz; Bruker, Billerica, MA, USA), a NMR system 500 MHz (^1^H, 500 MHz; ^13^C, 125 MHz, Varian, Palo Alto, CA, USA), using the solvent signals (δ_H_ 7.24/δ_C_ 77.00 for CDCl_3_; Cambridge Isotope Laboratories, Inc., Tewksbury, MA, USA) as internal standards; chemical shifts are indicated as δ values. Column chromatography was performed over silica gel 60 (70−230 mesh, Merck, Darmstadt, Germany). Silica gel 60 F254 and RP-18 F254S plates (Merck) were used for analysis by thin-layer chromatography (TLC) under detection of ultraviolet (UV) and 10% H_2_SO_4_ reagent to visualize the compounds. The analytical grade of solvents was used for the whole experiments.

### 3.2. Fungal Materials

AJF1 was isolated from the flowers of *Aconitum carmichaeli* collected from Jangbaek Mountain, Gangwon-do, South Korea, in 2016. The collected flowers were sterilized with 70% ethanol for 1 min and 7% H_2_O_2_ for 1 min, and then washed with sterilized distilled water. The sterilized flowers were cut into small pieces, and then diluted with sterilized water. The diluted solution was poured onto YME agar media (Yeast Ex, 4 g/L; Malt Ex, 10 g/L; Dextrose, 4 g/L; and sterilized water 1L), which was then incubated at 28 °C for 1 month. A pure strain of incarnadine colored colony (AJF1) was obtained and the strain AJF1 was identified to be *Aureobasidium pullulans* based on 16S/18S rDNA sequence analysis (97.72%, similarity to *Aureobasidium pullulans* strain YY20).

### 3.3. Extraction and Isolations

The yeast strain was inoculated in 36 L of YME media (Yeast Ex, 4 g/L; Malt Ex, 10 g/L; Dextrose, 4 g/L; and sterilized water 1L) at 25 °C for 10 days at 150 rpm. The cultured media of the strain (36 L) was extracted with EtOAc twice (36 L × 2) and the EtOAc layer was concentrated in vacuo to give crude extracts (7.9 g). The EtOAc extract (7.9 g) was subjected to vacuum liquid chromatography (VLC, 40 cm × 9 cm) over silica gel using stepwise gradient solvents of hexanes/acetone (10:0, 9:1. 8:2, 7:3, 6:4. 5:5, and 0:10; each 1 L), and 100% MeOH (1 L) to obtain fifteen fractions (fractions 21FA-21FO). Fraction 21FI (318.0 mg) was separated using sephadex LH-20 (50 cm × 2 cm) with elution of 100% MeOH to afford compounds **1** (39.7 mg), **2** (74.1 mg), and 3 (28.2 mg). Compounds **1**–**3** were further purified by chromatographic method over reverse-phased silica gel column (4 cm × 10 cm) using a H_2_O/MeOH (40:60→10:90, *v*/*v*, each 100 mL) gradient system. The spectral data for the isolated and reacted compounds were added as Appendix A.

#### 3.3.1. (3*R*,5*R*)-3,5-dihydroxydecanoic acid (**1**)

Colorless oil. [α${]}_{D}^{25}$ + 37.4 (*c* 0.1, MeOH); ^1^H-NMR (CDCl_3_, 500 MHz) δ 4.66 (1H, dddd, *J* = 11.0, 7.9, 5.3, 3.0 Hz, H-5), 4.33 (1H, q, *J* = 3.7 Hz, H-3), 2.67 (1H, dd, *J* = 17.7, 4.9 Hz, H-2a), 2.58 (1H, ddd, *J* = 17.7, 3.7, 1.7 Hz, H-2b), 1.93 (1H, *J* = dddd, 14.4, 3.8, 3.0, 1.7 Hz, H-4a), 1.68 (1H, ddd, *J* = 14.4, 11.0, 3.0 Hz, H-4b), 1.65 (1H, m, H-6b), 1.55 (1H, sep, *J* = 5.3 Hz, H-6a), 1.46 (1H, dddt, *J* = 10.2, 8.0, 5.3, 2.9 Hz, H-7a), 1.35 (1H, dddd, *J* = 12.7, 10.2, 8.0, 5.3, H-7b), 1.27 (2H, m, H_2_-9), 1.25 (2H, m, H_2_-8), 0.85 (3H, t, *J* = 6.9 Hz, H_3_-10); ^13^C-NMR (CDCl_3_, 125 MHz) *δ* 171.1 (C-1), 76.1 (C-5), 62.5 (C-3), 38.5 (C-2), 35.7 (C-4), 35.4 (C-6), 31.5 (C-8), 24.5 (C-7), 22.5 (C-9), 13.9 (C-10); HMBC correlations (CDCl_3_, H-# → C-#) H-5 → C-3, C-4, and C-7; H-3 → C-1, C-4, and C-5; H-2 → C-1, C-3, and C-4; H-4 → C-2 and C-3; H-6 → C-4, C-5, C-7, and C-8; H-7 → C-5, C-6, C-8, and C-9; H-8 → C-9 and C-10; H-8 → C-9; H-10 → C-8 and C-9; (+) HRFABMS obsd *m*/*z*, 187.1331 [M − H_2_O + H]^+^, calcd for C_10_H_19_O_3_, 187.1329.

#### 3.3.2. (3*R*,5*R*)-3-(((3*R*,5*R*)-3,5-dihydroxydecanoyl)oxy)-5-hydroxydecanoic acid (**2**)

Colorless oil. [α${]}_{D}^{25}$ + 1.8 (*c* 0.1, MeOH); ^1^H and ^13^C-NMR, see Table 1; HMBC correlations (CDCl_3_, H-# → C-#) H-3 → C-1, C-1’, C-2, C-4, and C-5; H-5 → C-3, C-4, C-6, and C-7; H-3′ → C-1′, C-2′, and C-5′; H-5′ → C-3′, C-4′, C-6′, and C-7′; H-2 → C-1, C-3, and C-4; H-2′ → C-1′, C-3′, and C-4′; H-4a → C-2 and C-3; H-4b → C-2, C-3, and C-5; H-6 → C-4, C-5, and C-7; CH_3_ → C-8, C-8′, C-9, and C-9′; (+) HRFABMS obsd *m*/*z*, 373.2590 [M − H_2_O + H]^+^, calcd for C_20_H_37_O_6_, 373.2585.

#### 3.3.3. (3*R*,5*R*)-3-(((3*R*,5*R*)-5-(((3*R*,5*R*)-3,5-dihydroxydecanoyl)oxy)-3-hydroxydecanoyl)oxy)-5-hydroxydecanoic acid (**3**)

Colorless oil. [α${]}_{D}^{25}$ − 5.1 (*c* 0.1, MeOH); ^1^H and ^13^C-NMR, see Table 1; HMBC correlations (CDCl_3_, H-# → C-#) H-3 → C-1, C-1′, C-2, C-4, and C-5; H-5′ → C-1″, C-3′, C-4′, C-6′ and C-7′; H-3″ → C-1″, C-2″, and C-5″; H-3′ → C-1′; H-5″ → C-4″, C-6″, and C-7″; H-2a and H-2b → C-1, C-3, and C-4; H-2′a and H-2′b → C-1′, C-3′, and C-4′; H-2″a and H-2″b → C-1″, C-3″, and C-4″; H-4a and H-4b → C-3 and C-5; H-4a′ and H-4b′ → C-3′ and 5′; H-4a″ and H-4b″ → C-3″ and 5″; (+) HRFABMS obsd *m*/*z*, 581.3657 [M − H_2_O + Na]^+^, calcd for C_30_H_54_O_9_Na, 581.3660.

### 3.4. Chemical Synthesis for Structure Determination

#### 3.4.1. Acid Hydrolysis of Compounds **2** and **3**

Each 10 mg of compounds **2** and **3** in methanol (1 mL) was stirred for 1 h at 60 °C after 2 mL of 1 N HCl was added. 1 N aqueous NaOH was added to the reaction mixture for neutralization, followed by extraction with EtOAc (3 mL), and the EtOAc extract was subjected to column chromatography over silica gel (4 cm × 10 cm) using a gradient system of hexane and acetone (2%, 4%, 6%, 8%, and 10%) to afford **2a** (3,5-dihydroxydecanoic acid methyl ester; hydrolysate of **2**), **2b** (3, 5-dihydroxydecanoic acid; hydrolysate of **2**), **3a** (3,5-dihydroxydecanoic acid methyl ester; hydrolysate of **3**), and **3b** (3,5-dihydroxydecanoic acid; hydrolysate of **3**).

**2a** and **3a**: ^1^H-NMR (300 MHz, CDCl_3_): δ 4.26 (1H, quintet, J = 6.3 Hz, H-3), 3.85 (1H, dd, J = 7.7, 4.1 Hz, H-5), 3.70 (3H, s, OCH_3_), 2.51 (2H, m, H_2_-2), 1.55 (2H, m, H-4 and 6a), 1.40 (2H, m, H-6b and 7), 1.28 (4H, m, H_2_-8 and 9), 0.89 (3H, t, J = 6.6 Hz, H_3_-10). ^13^C-NMR (75 MHz, CDCl_3_): δ 172.9 (C-1), 72.3 (C-3), 69.1 (C-5), 51.8 (OCH_3_), 42.2 (C-2), 41.5 (C-4), 37.8 (C-6), 31.8 (C-8), 25.0 (C-7), 22.6 (C-9), 14.0 (C-10).

**2b** and **3b**: ^1^H-NMR (300 MHz, CDCl_3_): δ 4.66 (1H, dtd, J = 11.3, 5.0, 3.8 Hz, H-5), 4.37 (1H, quintet, J = 3.8 Hz, H-3), 2.72 (1H, dd, J = 17.7, 5.0 Hz, H-2a), 2.61 (1H, ddd, J = 17.7, 3.8, 1.6, H-2b), 1.93 (1H, dtd, J = 14.4, 3.8, 1.6 Hz, H-4a), 1.76 (1H, ddd, J = 14.4, 11.3, 3.8 Hz, H-4b), 1.68 (1H, m, H-6a), 1.55 (1H, m, H-6b), 1.46 (1H, m, H-7a), 1.35 (1H, m, H-7b), 1.31 (2H, m, H-8), 1.24 (2H, m, H-9), 0.92 (3H, t, J = 6.7 Hz, H-10).

#### 3.4.2. Synthesis of 1,3-diol-acetonides for Compounds **1**, **2**, and **3**

Each of the purified compounds **1, 2a**, and **2b** as well as and mixture of **3a** and **3b** (5 mg) were dissolved in 2,2-dimethoxypropane (5 mL) and methanol (1 mL), followed by addition of pyridinium-*p*-toluenesulfonate (5 mg). The mixture was stirred for 12 h at 25 °C for the reaction. After quenching with 5% aqueous NaHCO_3_, the reaction mixture was extracted with CH_2_Cl_2_ three times. The CH_2_Cl_2_ soluble fraction was evaporated *in vacuo* and further purified by silica gel column (4 cm × 10 cm) using a gradient solvent system of n-hexane and acetone (2%, 4%, 6%, 8%, and 10%) to afford **1a** (acetonide product of **1**), **2c** (acetonide product of **2a**), **2d** (acetonide product of **2b**), and **3c** (acetonide product of **3a** and **3b**).

Acetonide derivatives of **1**, hydrolysates of **2** and **3** (**1a**, **2c**, and **3c**)

^1^H-NMR (300 MHz, CDCl_3_): δ 4.27 (1H, dtd, *J* = 11.6, 6.5, 2.5 Hz, H-3), 3.79 (1H, m, H-5), 3.66 (3H, s, OCH_3_), 2.53 (1H, dd, *J* = 15.6, 6.8 Hz, H-2a), 2.36 (1H, dd, *J* = 15.6, 6.2 Hz, H-2b), 1.57 (1H, dt, *J* = 12.6, 2.4 Hz, H-4a), 1.42 (3H, s, CH_3_), 1.34 (3H, s, CH_3_), 1.25 (6H, m), 0.90 (3H, t, *J* = 6.8 Hz, H_3_-10). ^13^C-NMR (75 MHz, CDCl_3_): δ 171.5 (C-1), 98.7 (OCO), 68.8 (C-3), 65.9 (C-5), 51.6 (OCH_3_), 41.3 (C-2), 36.5 (C-4), 36.3 (C-6), 31.8 (C-8), 30.1 (CH_3_), 24.6 (C-7), 22.6 (C-9), 19.7 (CH_3_), 14.0 (C-10).

#### 3.4.3. Deprotection of 1,3-diol-acetonide for Compound **3c**

Acetonide product (**3c**) in MeOH (2 mL) was stirred for 2 h at 40 °C for the reaction after adding pyridinium-*p*-toluenesulfonate (5 mg). The reaction mixture was extracted with CH_2_Cl_2_ two times. The CH_2_Cl_2_ soluble fraction was evaporated in vacuo and further purified by silica gel column (4 cm × 10 cm) using a gradient solvent system of n-hexane and acetone (2%, 4%, 6%, 8%, and 10%) to afford **3d.**

#### 3.4.4. Mosher’s Esterification of **1**, **2**, and **3d**

Mosher’s esterifications of compounds **1**, **2**, and **3d** were performed in NMR solvents using (*S*)- and (*R*)-MTPA [α-methoxy-α-(trifluoromethyl)phenyl-acetyl] chloride. Each compounds **1**, **2**, and **3d** (1 mg) in pyridine-*d*_5_ (0.5 mL) were treated with (*R*)-MTPA-Cl or (*S*)-MTPA-Cl (6.0 μL) and 4-(dimethylamino)-pyridine (one crystal). The vials containing reaction mixtures were immediately sealed and shaken for the reaction. Treatments of each compound with (*R*)-MTPA-Cl afforded (*S*)-MTPA ester, and vice versa. After being put at room temperature for one hour, the solutions were moved to NMR tube then monitored by ^1^H-NMR.

(*S*)-MTPA ester of **1**: ^1^H-NMR (500 MHz, Pyridine-d_5_): δ 5.776 (1H, quintet, J = 4.1 Hz, H-3), 4.562 (ddt, J = 11.5, 7.6, 3.7 Hz, H-5), 3.239 (1H, dd, J = 17.8, 5.4 Hz, H-2a), 3.029 (1H, dd, J = 17.8, 3.8 Hz, H-2b), 2.228 (1H, d, J = 15.0 Hz, H-4a), 2.009 (1H, ddd, J = 15.0, 11.3, 3.6 Hz, H-4b), 1.554 (1H, m), 1.439 (1H, m), 1.342 (1H, m), 1.201 (m), 1.103 (m), 0.747 (3H, t, J = 7.1 Hz, H_3_-10).

(*R*)-MTPA ester of **1**: ^1^H-NMR (500 MHz, Pyridine-*d*_5_): δ 5.764 (1H, quintet, *J* = 3.9 Hz, H-3), 4.305 (1H, ddt, 11.6, 7.7, 3.6 Hz, H-5), 3.255 (1H, dd, *J* = 17.9, 5.3 Hz, H-2a), 3.119 (1H, dd, *J* = 17.9, 3.8 Hz, H-2b), 2.111 (1H, d, *J* = 15.0 Hz, H-4a), 1.927 (1H, ddd, *J* = 15.0, 11.4, 3.3 Hz, H-4), 1.500 (1H, m), 1.366 (1H, m), 1.278 (m), 1.124 (m, 1.065 (m), 0.756 (3H, t, *J* = 7.1 Hz, H_3_-10).

(*S*)-MTPA ester of **2**: ^1^H-NMR (500 MHz, Pyridine-d_5_): δ 5.957 (1H, ddq, J = 9.7, 6.6, 3.5 Hz, m, H-3), 5.497 (1H, quintet, J = 6.3 Hz, H-5), 5.378 (1H, dq, J = 5.5, 3.6 Hz, H-3′), 4.628 (1H, ddd, J = 15.1, 7.3, 3.7 Hz, H-5′), 3.132 (1H, dd, J = 16.8, 3.5 Hz, H-2a), 3.049 (1H, dd, J = 17.7, 5.5 Hz, H-2a′), 2.904 (1H, dd, J = 16.8, 9.0 Hz, H-2b), 2.800 (1H, ddd, J = 17.9, 3.7, 1.6 Hz, H-2b′), 2.471 (1H, dt, J = 14.4, 6.5 Hz, H-4a), 2.253 (1H, ddd, J = 14.3, 7.2, 5.7 Hz, H-4b), 2.100 (1H, d, J = 14.0, H-4a′), 1.859 (1H, ddd, J = 14.9, 11.6, 3.6 Hz, H-4b′), 1.600 (1H, m, 1.473 (1H, m), 1.423 (1H, m), 1.196 (m), 0.814 (6H, m).

(*R*)-MTPA ester of **2**: ^1^H-NMR (500 MHz, Pyridine-d_5_): δ 6.017 (1H, m, H-3), 5.480 (1H, quintet, J = 3.9 Hz, H-5), 5.374 (1H, dq, J = 12.0, 6.5 Hz, H-3’), 4.682 (1H, ddd, J = 11.1, 7.6, 3.5 Hz, H-5’), 3.319 (1H, dd, J = 16.8, 3.6 Hz, H-2a), 3.153 (1H, d, J = 5.4 Hz, H-2a′), 3.039 (1H, dd, J = 16.5, 3.8 Hz, H-2b), 2.948 (1H, dd, J = 16.9, 3.0 Hz, H-2b′), 2.433 (1H, dt, J = 13.8, 6.6 Hz, H-4a), 2.191 (1H, dt, J = 13.8, 6.6 Hz, H-4b), 2.121 (1H, d, J = 14.9 Hz, H-4a′), 1.876 (1H, ddd, J = 14.9, 11.4, 3.5 Hz, H-4b′), 1.612 (1H, m), 1.468 (1H, m), 1.306 (1H, m), 1.156 (m), 0.821 (3H, t, J = 6.9 Hz), 0.785 (3H, t, J = 6.8 Hz).

(*S*)-MTPA ester of **3d**: ^1^H-NMR (500 MHz, Pyridine-d_5_): δ 5.899 (1H, td, J = 10.0, 5.3 Hz, H-3), 5.481 (1H, quintet, J = 6.5 Hz, H-5), 3.309 (1H, dd, J = 16.5, 3.8 Hz, H-2a), 2.449 (1H, dd, J = 13.8, 6.5 Hz, H-4a), 2.220 (1H, dt, J = 13.8, 6.5 Hz, H-4b), 1.810 (2H, m), 0.812 (3H, t, J = 6.5 Hz, H-10).

(*R*)-MTPA ester of **3d**: ^1^H-NMR (500 MHz, Pyridine-d_5_): δ 5.983 (1H, ddt, J = 10.2, 6.3, 3.8 Hz, H-3), 5.374 (1H, quintet, J = 6.3 Hz, H-5), 3.195 (1H, dd, J = 16.5, 4.0 Hz, H-2a), 2.415 (1H, dt, J = 13.8, 6.8 Hz, H-4a), 2.166 (1H, dt, J = 13.8, 6.8 Hz, H-4b), 1.642 (2H, m), 0.781 (3H, t, J = 6.5 Hz, H-10).

### 3.5. Biological Activity Evaluation

#### 3.5.1. Antibacterial Assay

The antibacterial activity was evaluated by the broth microdilution method [19]. Four bacterial strains (*Escherichia coli* ATCC 11775, *Staphylococcus aureus* ATCC 6538, *Bacillus subtilis* ATCC 6633, and *Pseudomonas aeruginosa*) were cultured on Mueller-Hinton (MH) agar plates for 24 h at 24 °C (for *B. subtilis* ATCC 6633) or 37 °C (for three other strains). The bacterial colonies were suspended in MH broth and cultured for 24 h at 24 or 37 °C with 220 rpm. The bacterial suspension was diluted with MH broth with 0.5 McFarland turbidity equivalents. Gentamicin (128 μg, positive control) and the compounds **1**–**3** (128 μg, respectively) were dissolved in DMSO (12 μL) and diluted with 488 μL of MH broth to prepare stock solution. One hundred microliter of the stock solution were transferred to a well of 96-well plate and serially diluted by using two-fold dilutions. The bacterial suspension (50 μL) was added to each well and the concentrations of test compounds were 128 μg/mL. The 96-well plate was incubated for 16 h at 24 or 37 °C.

#### 3.5.2. Cytotoxicity

MCF-7 human breast cancer cell line was obtained from American Typr Culture Collection (ATCC, Manassas, VA, USA) and cultured in RPMI1640 medium (Cellgro, Manassas, VA, USA) supplemented with 10% FBS (Gibco BRL, Grand Island, NY, USA), 100 μg/mL streptomycin, and 100 U/mL penicillin. Cultures were maintained at 37 °C in a humidified incubator with 95% air and 5% CO_2_. The Ez-Cytox cell viability detection kit (Daeil Lab Service Co., Seoul, South Korea) was used for quantification of cell viability. MCF-7 cells were seeded in 96-well plates (1 × 10^4^ cells per well) for 24 h and treated for the indicated concentrations of test samples in RPMI1640 medium. The Ez-Cytox reagents were added to each well and the absorbance values at 450 nm was measured using a microplate reader (PowerWave XS; Bio-Tek Instruments, Winooski, VT, USA).

## 4. Conclusions

An endophytic yeast, *Aureobasidium pullulans* AJF1, was isolated from flowers of the *Aconitum carmichaeli* in this study and cultivated on a large scale for chemical investigation. Extensive chemical investigation of the extracts resulted in isolation of three lipid type compounds (**1**–**3**). They were (3*R*,5*R*)-3,5-dihydroxydecanoic acid unit or the esterified complexes of the unit. Relative and absolute stereochemistries of the isolated compounds (**1**–**3**) were established using modified Mosher’s method together with analysis of NMR data for their acetonide derivatives. Even though compounds **2** and **3** were all unique lipid type new compounds, they did not show potent antibiotic activities against *Escherichia coli*, *Staphylococcus aureus*, *Bacillus subtilis*, *Pseudomonas aeruginosa,* and no cytotoxicities against MCF-7 cancer cells. Regardless of their low potencies of activities, it is a good finding that endophytes from medicinal plants could be good sources for new chemistries.

## Figures and Tables

**Figure 1 molecules-24-04051-f001:**
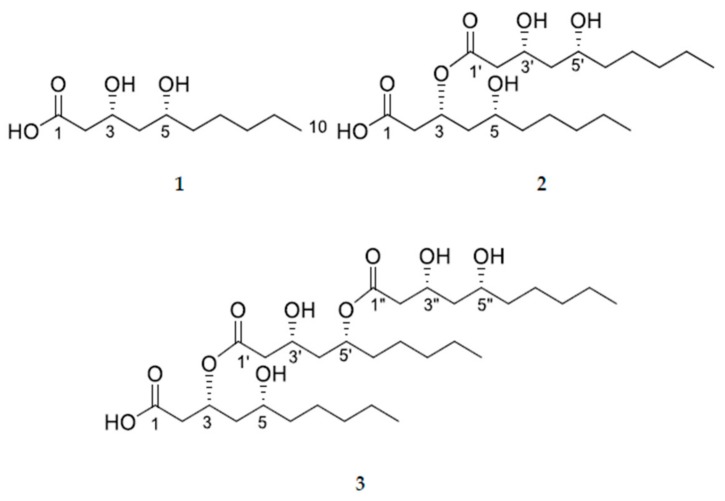
Structures of (3*R*,5*R*)-3,5-dihydroxydecanoic acid (**1**), (3*R*,5*R*)-3-(((3*R*,5*R*)-3,5-dihydroxydecanoyl)oxy)-5-hydroxydecanoic acid (**2**), and (3*R*,5*R*)-3-(((3*R*,5*R*)-5-(((3*R*,5*R*)-3,5-dihydroxydecanoyl)oxy)-3-hydroxydecanoyl)oxy)-5-hydroxydecanoic acid (**3**).

**Figure 2 molecules-24-04051-f002:**
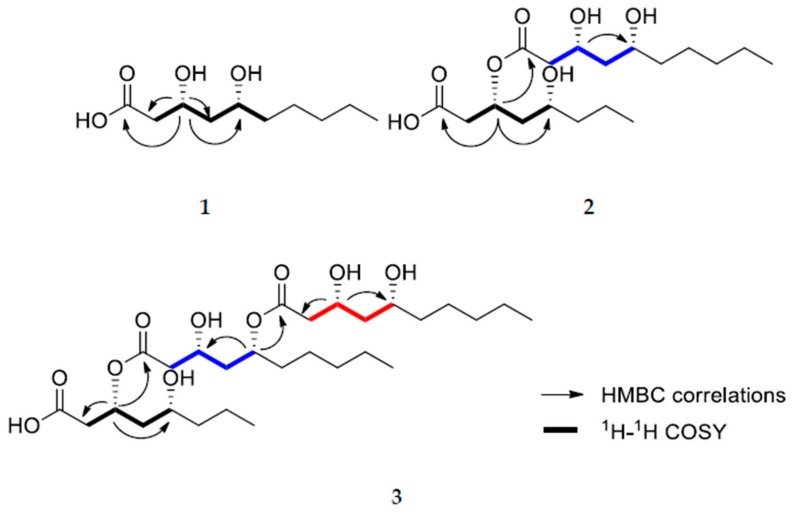
Key HMBC correlations for compounds **1**–**3** and ^1^H-^1^H COSY correlations.

**Figure 3 molecules-24-04051-f003:**
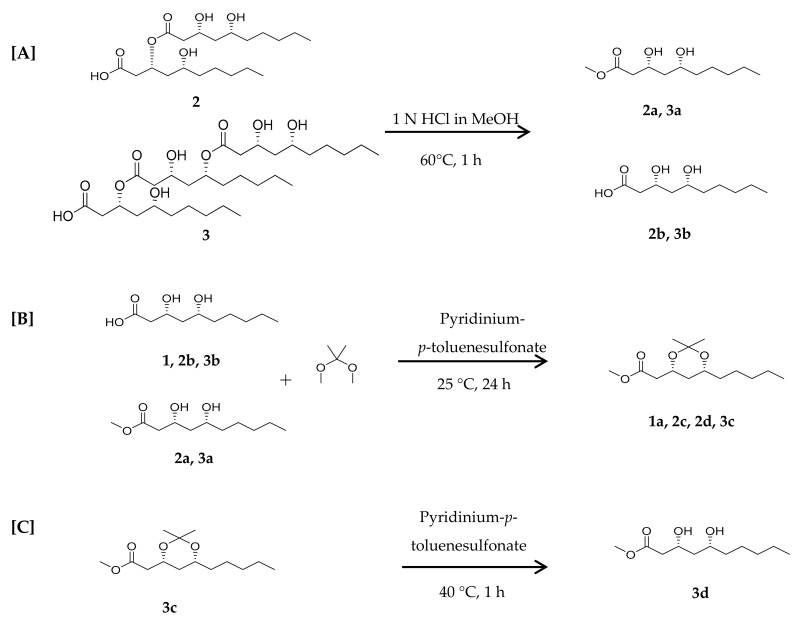
Reaction procedures for elucidation of stereochemistry for compounds **1**–**3**; (**A**) hydrolysis of compounds **2** and **3**; (**B**) synthesis of 1,3-diol-acetonides for **1**, **2b**, **3b**, **2a**, and **3a**; and (**C**) deprotection of acetonide for **3c**.

**Figure 4 molecules-24-04051-f004:**
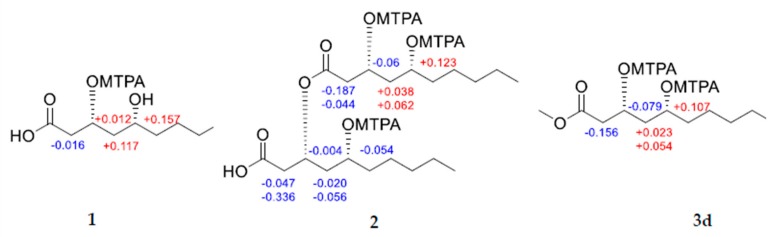
Observed chemical shift differences (∆δ*_S-R_* = δ*_S_* − δ*_R_*, in ppm, 500 MHz) for the *R*- and *S*-MTPA esters of compounds **1**, **2**, and **3d**.

**Table 1 molecules-24-04051-t001:** ^1^H and ^13^C-NMR data of **2** and **3** (δ values).^a^^)^

No.	2		3	
	δ_H_ (m, *J* in Hz) ^b)^	δ_C_ ^c)^	δ_H_ (m, *J* in Hz) ^b)^	δ_C_ ^c)^
1		169.2		169.0
2a2b	2.77 (dd, *J* = 18.1, 5.4 Hz) 2.69 (ddd, *J* = 18.1, 3.5, 1.6 Hz)	35.3	2.76 (dd, 18.1, 5.3), 2.68 (ddd, 18.1, 3.5, 1.6)	35.3
3	5.27 (dt, *J* = 7.0, 3.5 Hz)	65.7	5.26 (dq, 7.2, 3.5)	66.1
4a4b	2.07 (dtd, *J* = 14.9, 3.5, 1.6 Hz) 1.77 (ddd, *J* = 14.9, 11.5, 3.5 Hz)	32.7	2.07 (dtd, 14.9, 3.5, 1.6) 1.76 (ddd, 14.9, 12.3, 4.0)	32.8
5	4.53 (dtd, *J* = 11.5, 5.0, 2.9 Hz)	76.1	4.53 (dtd, 11.1, 4.9, 3.1)	76.2
6a6b	1.69 (dddd, *J* = 13.5, 10.2, 7.5, 5.0 Hz) 1.57 (m)	35.2	0.55 (m)	34.7
7a7b	1.46 (m) 1.37 (m)	24.3	1.47 (m) 1.36 (m)	24.4
8	1.27 (m)	31.7	1.25 (m)	31.8
9	1.35 (m)	22.5	1.35 (m)	22.5
10	0.86 (t, *J* = 6.9 Hz)	13.9	0.86 (t, 6.9)	14.0
1′		171.1		171.5
2′a2′b	2.48 (m) 2.48 (m)	42.2	2.52 (dd, 16.1, 3.6) 2.45 (m)	41.7
3′	4.25 (ddt, 9.4, 7.5, 5.1)	68.9	4.10 (tt, 8.2, 4.0)	66.3
4′a4′b	1.56 (m) 1.54 (m)	42.0	1.83 (m) 1.64 (m)	41.0
5′	3.85 (ddd, 12.1, 7.5, 4.5)	72.3	5.04 (tt, 8.9, 4.8)	72.9
6′a6′b	1.45 (m) 1.41 (m)	37.8	1.68 (m) 1.57 (m)	35.4
7′a7′b	1.46 (m) 1.37 (m)	24.9	1.47 (m) 1.36 (m)	24.8
8′	1.25 (m)	31.4	1.25 (m)	31.5
9′	1.27 (m)	22.4	1.35 (m)	22.5
10′	0.86 (t, 6.9)	13.9	0.86 (t, 6.9)	14.0
1′′				172.3
2′′a2′′b			2.45 (m) 2.40 (m)	42.7
3′′			4.25 (tt, 8.5, 4.0)	69.8
4′′a4′′b			1.55 (m)	42.4
5′′			3.84 (tt, 8.1, 4.1)	72.4
6′′a6′′b			1.44 (m) 1.55 (m)	37.9
7′′a7′′b			1.37 (m) 1.26 (m)	25.0
8′′			1.25 (m)	31.5
9′′			1.27 (m)	22.5
10′′			0.86 (t, 6.9)	14.0

^a)^ δ value recorded in ppm from TMS (tetramethylesilane) at room temperature. ^b)^
^1^H (500 MHz) and ^c)^
^13^C (125 MHz) NMR measured in CDCl_3_.

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
