# Peer review of "New Hydroxydecanoic Acid Derivatives Produced by an Endophytic Yeast Aureobasidium pullulans AJF1 from Flowers of Aconitum carmichaeli"

_molecules, 2019, doi:10.3390/molecules24224051_

Round 1

Reviewer 1 Report

The paper by Hyun Gyu Choi et all is potentially interesting. However, I have some big concerns
introduction, discussion, and bibliography are poor
the isolated compounds seem to have no antimicrobial effect, cytotoxic activity, etc.
No data about the biological activity (even if low) of these compounds,
There are not suggestions about the application of these molecules
the importance of the characterization of these compounds is not clear

Author Response

Answers of Reviewers’ Comments

Manuscript ID: molecules-628188

Title: "Three new hydroxydecanoic acid derivatives produced by an endophytic yeast Aureobasidium pullulans AJF1 from flowers of Aconitum carmichaeli"

Authors: Hyun Gyu Choi, Jung Wha Kim, Hyukjae Choi and Sang Hee Shim

Dear editor,

Thank you very much for your careful review on our manuscript.

According to reviewers’ comments, all comments were answered one by one as shown below. Thus, several parts of the revised manuscript have been corrected and indicated in blue color concerning some modification in comparison with the previous manuscript.

------------------------------------------------------------------------------------------------------------------------

Reviewer #1

The paper by Hyun Gyu Choi et al. is potentially interesting. However, I have some big concerns introduction, discussion, and bibliography are poor. The isolated compounds seem to have no antimicrobial effect, cytotoxic activity, etc. No data about the biological activity (even if low) of these compounds,
There are not suggestions about the application of these molecules
the importance of the characterization of these compounds is not clear.

Answer: Thanks for the careful review. According to the reviewer’s comments, we added more information in the parts of introduction, discussion and bibliography. And we added the details on the bioactivities in the experimental section. Even though the isolated compounds did not show antibacterial activities, we report on the isolation and identification of two new lipid-type compounds produced by an endophyte, which was isolated from roots of Aconitum carmichaeli, which has a pharmacological importance as a traditional medicine. Furthermore, we employed many scientific chemical and spectral methods to elucidate relative and absolute stereochemistries of the isolated lipid type compounds.

-------------------------------------------------------------------------------------------------------------------

If you need further information or more information, please let me know.

Best regards,

Sang Hee Shim

Professor

College of Pharmacy, Duksung Women’s University

144Gil-33, Samyang-ro, Dobong-gu, Seoul 01369, South Korea

Reviewer 2 Report

Very nice study, well conducted, which deserve to be published. The methodology is well described and the compounds isolated and characterized could be of interest for people working in the field of natural product and pharmaceutical chemistry.

Some minor typos errors were detected:

line 79 "and H NMR data" remove the H

line 80 "in room temperature" change by "at room temperature"

Author Response

Answers of Reviewers’ Comments

Manuscript ID: molecules-628188

Title: "Three new hydroxydecanoic acid derivatives produced by an endophytic yeast Aureobasidium pullulans AJF1 from flowers of Aconitum carmichaeli"

Authors: Hyun Gyu Choi, Jung Wha Kim, Hyukjae Choi and Sang Hee Shim

Dear editor,

Thank you very much for your careful review on our manuscript.

According to reviewers’ comments, all comments were answered one by one as shown below. Thus, several parts of the revised manuscript have been corrected and indicated in blue color concerning some modification in comparison with the previous manuscript.

-----------------------------------------------------------------------------------------

Reviewer #2

Very nice study, well conducted, which deserve to be published. The methodology is well described and the compounds isolated and characterized could be of interest for people working in the field of natural product and pharmaceutical chemistry.

Answer: Thanks for the review and good comments.

Some minor typos errors were detected:

line 79 "and H NMR data" remove the H

→ According to the comment, H was removed from the “H NMR data”.

line 80 "in room temperature" change by "at room temperature"

→ According to this comment, “in” was replaced by “at”.

If you need further information or more information, please let me know.

Best regards,

Sang Hee Shim

Professor

College of Pharmacy, Duksung Women’s University

144Gil-33, Samyang-ro, Dobong-gu, Seoul 01369, South Korea

Reviewer 3 Report

Three new hydroxydecanoic acid derivatives produced by an endophytic yeast Aureobasidium pullulans

AJF1 from flowers of Aconitum carmichaeli

Summary:

This article details the isolation and structure elucidation of 3 compounds from the endophytic yeast Aureobasidium pullulans. The structures were identified using rigorous techniques, and the science of the paper is outstanding. However, I am somewhat concerned that it seems like one of the compounds they describe as being new has been previously isolated from a natural source (see suggestion 1). The authors could leverage some of the known compounds with similar structures to discuss structure-activity relationship of this class of compounds, but that was not done in this manuscript (see suggestions 2 & 5). Finally, there were some formatting issues both in the main text and the supplemental material that should be addressed prior to publication (see suggestions 3 & 4). With that being said, I maintain that the science is excellent, and there should be some interest in these molecules based both on their similarity to known active compounds, and their ecological niche (medicinal plant endophyte). Therefore, I recommend accepting with major revisions.

Suggestions:

Upon a literature search, it appears that compound 1 has been reported previously as a natural product [Kurosawa, et. al. Extracellular Accumulation of the Polyol Lipids, 3,5- Dihydroxydecanoyl and 5-Hydroxy-2-decenoyl Esters of Arabitol and Mannitol, by Aureobasidium Biosci. Biotech. Biochem., 58 (II), 2057~2060, 1994]. This should acknowledged, and the manuscript should be modified to reflect this. In the introduction paragraph, the previous report of 3,5-dihydroxydecanoic acid derivatives from pullulans is not mentioned, even though a paper describing them is cited (the exophilins in Price, et. al.). I don’t understand the format of Table 1. It seems like there are many missing signals for 2. Maybe we are supposed to infer that the signals for 2 are identical to those of 3 for the first dihydroxydecanoic acid moiety, but that isn’t clear from the layout, and it looks like all the carbon signals for 3 are omitted, which would be strange, especially since you might expect a shift in 5’. The HMBC spectra in the SI have the intensity turned up so high that there is a lot of t1 noise. I understand that there are signals there that require this high intensity to show up, but it is also causing there to be many misleading apparent correlations (e.g., many around 120 and 130 ppm in Fig. 11). Maybe make two figures for the HMBC spectra, one with lower T1 noise, and one zoomed in on the weak signals of importance. Figure 11 in the SI must be changed, if anything it calls the purity or identity of 2 into doubt as currently shown with strong correlations to carbons not in the proposed Some discussion comparing structure and bioactivity of 3 to Exophilin A is warranted. Exophilin A showed activity against Gram-positive bacteria, including MRSA, while 3, which is only

structurally different based on location of esterification, was reported as inactive. Do the authors feel that this linkage is therefore important, or did their testing not go to high enough MIC values to see the activity, etc.? Also, an experimental section detailing how the bioassays were run should be included since their results are mentioned.

Suggested edits (page #, line #):

(1, 11) -- Remove the word “Recently” and start with “Endophytes…” (1, 13) -- “plants to produce…” to “plants that produce…”

(1, 16) -- “flower of the Aconitum…” to “flower of Aconitum…” (1, 22) -- “Stereochemistries…” to “Absolute configurations…” (1, 30) -- “whole” to “all”

(1, 30) -- “tissues of host…” to “tissues of a host…”

(1, 31) -- “It has been reported to cause…” to “They cause…”

(1, 33) -- The sentence starting “There reported various types of secondary metabolites…” should start “Various types of secondary metabolites…”

(1, 36) -- circumstance is not an environmental signal

(1, 42) -- “drugs in the oriental medicine…” to “drugs in oriental medicine…” (1, 43) -- citation needed for the sentence starting “Roots of Aconitum…” (2, 46) -- “researches” to “studies”

(2, 57) -- “cultures of the endophytic yeast strain of…” to “cultures of an endophytic strain of…” (2, 65) -- the calculated m/z value for the molecular formula is not shown

(2, 72) -- “Furthermore,” to “A”

(2, 73) -- Change “HMBCs” to “HMBC correlations” here and throughout the remainder of the manuscript.

(4, 94) -- “…was out of question…” to “…was confirmed…”

***No further edits due to time constraints

Author Response

Answers of Reviewers’ Comments

Manuscript ID: molecules-628188

Title: "Three new hydroxydecanoic acid derivatives produced by an endophytic yeast Aureobasidium pullulans AJF1 from flowers of Aconitum carmichaeli"

Authors: Hyun Gyu Choi, Jung Wha Kim, Hyukjae Choi and Sang Hee Shim

Dear editor,

Thank you very much for your careful review on our manuscript.

According to reviewers’ comments, all comments were answered one by one as shown below. Thus, several parts of the revised manuscript have been corrected and indicated in blue color concerning some modification in comparison with the previous manuscript.

Reviewer #3

Summary:

This article details the isolation and structure elucidation of 3 compounds from the endophytic yeast Aureobasidium pullulans. The structures were identified using rigorous techniques, and the science of the paper is outstanding. However, I am somewhat concerned that it seems like one of the compounds they describe as being new has been previously isolated from a natural source (see suggestion 1). The authors could leverage some of the known compounds with similar structures to discuss structure-activity relationship of this class of compounds, but that was not done in this manuscript (see suggestions 2 & 5). Finally, there were some formatting issues both in the main text and the supplemental material that should be addressed prior to publication (see suggestions 3 & 4). With that being said, I maintain that the science is excellent, and there should be some interest in these molecules based both on their similarity to known active compounds, and their ecological niche (medicinal plant endophyte). Therefore, I recommend accepting with major revisions.

Answer: Thanks for the careful review and good suggestions. Please refer to the following answers regarding each suggestion.

Suggestions:

Upon a literature search, it appears that compound 1 has been reported previously as a natural product [Kurosawa, et. al. Extracellular Accumulation of the Polyol Lipids, 3,5- Dihydroxydecanoyl and 5-Hydroxy-2-decenoyl Esters of Arabitol and Mannitol, by Aureobasidium Biosci. Biotech. Biochem., 58 (II), 2057~2060, 1994]. This should acknowledged, and the manuscript should be modified to reflect this. In the introduction paragraph, the previous report of 3,5-dihydroxydecanoic acid derivatives from pullulans is not mentioned, even though a paper describing them is cited (the exophilins in Price, et. al.).

→ According to this comment, we described the previous report of compound 1 in the introduction part (lines 48-51) and accordingly we reflected this report throughout the manuscript. Even though compound 1 was previously reported before, its relative and absolute stereochemistry was elucidated in this study for the first time.

I don’t understand the format of Table 1. It seems like there are many missing signals for 2. Maybe we are supposed to infer that the signals for 2 are identical to those of 3 for the first dihydroxydecanoic acid moiety, but that isn’t clear from the layout, and it looks like all the carbon signals for 3 are omitted, which would be strange, especially since you might expect a shift in 5’.

→ We would apologize for missing many important signals in Table 1. We added the missing signals for 2 and added all the carbon resonances for 3 in Table 1. In addition, the proton and carbon resonances for 1 were removed from Table 1 since it is already reported before and instead moved to the experimental parts (lines 217-223).

The HMBC spectra in the SI have the intensity turned up so high that there is a lot of t1 noise. I understand that there are signals there that require this high intensity to show up, but it is also causing there to be many misleading apparent correlations (e.g., many around 120 and 130 ppm in Fig. 11). Maybe make two figures for the HMBC spectra, one with lower T1 noise, and one zoomed in on the weak signals of importance. Figure 11 in the SI must be changed, if anything it calls the purity or identity of 2 into doubt as currently shown with strong correlations to carbons not in the proposed.

→ According to this comment, we made two figures for HMBC for compound 2 with ower T1 and one zoomed in as shown in the supporting materials.

Some discussion comparing structure and bioactivity of 3 to Exophilin A is warranted. Exophilin A showed activity against Gram-positive bacteria, including MRSA, while 3, which is only structurally different based on location of esterification, was reported as inactive. Do the authors feel that this linkage is therefore important, or did their testing not go to high enough MIC values to see the activity, etc.?

→ As the reviewer commented, compound 3 is different from exophilin A in the position of esterification. While exophilin A was reported to be active against G(+) bacteria, compound 3 did not show antibacterial activities even at 128 μg/mL against Escherichia coli ATCC 11775, Staphylococcus aureus ATCC 6538, Bacillus subtilis ATCC 6633, and Pseudomonas aeruginosa in this study. The position of the esterification might be or not be critical for the bioactivity. By the way, the activity of exophilin A against Staphyococcus aureus (the only pathogen tested for both exophilin A and compound 3 in common) was not potent with the MIC of 50 μg/mL.

Also, an experimental section detailing how the bioassays were run should be included since their results are mentioned.

→ According to this comment, we added the details on bioassays in the experimental sections.

Suggested edits (page #, line #):

(1, 11) -- Remove the word “Recently” and start with “Endophytes…” (1, 13) -- “plants to produce…” to “plants that produce…”

(1, 16) -- “flower of the Aconitum…” to “flower of Aconitum…” (1, 22) -- “Stereochemistries…” to “Absolute configurations…” (1, 30) -- “whole” to “all”

(1, 30) -- “tissues of host…” to “tissues of a host…”

(1, 31) -- “It has been reported to cause…” to “They cause…”

(1, 33) -- The sentence starting “There reported various types of secondary metabolites…” should start “Various types of secondary metabolites…”

(1, 36) -- circumstance is not an environmental signal

(1, 42) -- “drugs in the oriental medicine…” to “drugs in oriental medicine…” (1, 43) -- citation needed for the sentence starting “Roots of Aconitum…” (2, 46) -- “researches” to “studies”

(2, 57) -- “cultures of the endophytic yeast strain of…” to “cultures of an endophytic strain of…” (2, 65) -- the calculated m/z value for the molecular formula is not shown

(2, 72) -- “Furthermore,” to “A”

(2, 73) -- Change “HMBCs” to “HMBC correlations” here and throughout the remainder of the manuscript.

(4, 94) -- “…was out of question…” to “…was confirmed…”

→ According to these comments, we revised all the points the reviewer commented in the manuscript.

If you need further information or more information, please let me know.

Best regards,

Sang Hee Shim

Professor

College of Pharmacy, Duksung Women’s University

144Gil-33, Samyang-ro, Dobong-gu, Seoul 01369, South Korea

Round 2

Reviewer 1 Report

the authors answered all my requests 

Author Response

Dear Reviewer of molecules,

We appreciate for your review and kind comments.

Sang Hee Shim

Reviewer 3 Report

All of the concerns that I expressed after the first review have been addressed.  However, upon further inspection of figure 11 in the supplemental information, it appears that what I initially thought was noise based on signal intensity being set too high, is actually a mirroring effect in the spectra.  This is what makes it appear as if there are correlations to carbons that don't exist in the structure.  I think this is a processing error, and should be easily corrected.

My only new suggestion is to change the title from "Three new hydroxydecanoic acid..." to "New hydroxydecanoic acid...", since only two would be considered new.

Author Response

Dear Reviewer,

We appreciate for your careful review.

As the reviewer pointed out, there is a mirroring effects in the HMBC spectrum for compound 2. Herein, we attached the HMBC data with adjustment of T1 noise which was taken in 300 MHz as attached. This spectrum was added as a Fig. 11(b) on the supplementary file.

In addition, according to the suggestion, we revised the title from "Three new hydroxydecanpoic acid..." to "New hydroxydecanoic acid...".

Thank you,

Sang Hee Shim